# A Lightweight Algorithm for Insulator Target Detection and Defect Identification

**DOI:** 10.3390/s23031216

**Published:** 2023-01-20

**Authors:** Gujing Han, Liu Zhao, Qiang Li, Saidian Li, Ruijie Wang, Qiwei Yuan, Min He, Shiqi Yang, Liang Qin

**Affiliations:** 1Department of Electronic and Electrical Engineering, Wuhan Textile University, Wuhan 430200, China; 2State Key Laboratory of New Textile Materials and Advanced Processing Technologies, Wuhan Textile University, Wuhan 430200, China; 3State Grid Information & Telecommunication Group Co., Ltd., Beijing 102211, China; 4School of Electrical and Automation, Wuhan University, Wuhan 430072, China

**Keywords:** lightweight algorithm, YOLOv4, GhostNet, insulator, attention mechanism

## Abstract

The accuracy of insulators and their defect identification by UAVs (unmanned aerial vehicles) in transmission-line inspection needs to be further improved, and the model size of the detection algorithm is significantly reduced to make it more suitable for edge-end deployment. In this paper, the algorithm uses a lightweight GhostNet module to reconstruct the backbone feature extraction network of the YOLOv4 model and employs depthwise separable convolution in the feature fusion layer. The model is lighter on the premise of ensuring the effect of image information extraction. Meanwhile, the ECA-Net channel attention mechanism is embedded into the feature extraction layer and PANet (Path Aggregation Network) to improve the recognition accuracy of the model for small targets. The experimental results show that the size of the improved model is reduced from 244 MB to 42 MB, which is only 17.3% of the original model. At the same time, the mAp of the improved model is 0.77% higher than that of the original model, reaching 95.4%. Moreover, the mAP compared with YOLOv5-s and YOLOX-s, respectively, is improved by 1.98% and 1.29%. Finally, the improved model is deployed into Jetson Xavier NX and run at a speed of 8.8 FPS, which is 4.3 FPS faster than the original model.

## 1. Introduction

Insulators are important components in transmission lines, playing the role of mechanical support and line insulation [1]. Since insulators are in the outdoor natural environment for a long time, they are prone to self-destruction and defect due to temperature change, moisture and lightning strike, etc. Therefore, target detection and defect identification of insulators are important guarantees for the safe and stable operation of transmission lines [2]. Traditional methods include manual inspection, traditional image-detection methods [3], etc. Manual inspection is dangerous and inefficient; traditional image processing mostly combines edge detection, color features and other methods, which are less accurate for insulator defect detection [4,5,6].

In recent years, deep learning algorithms have been developed for their efficiency and convenience in the application of UAV inspection of power transmission lines [7]. Deep learning algorithms for insulator target detection and defect recognition based on aerial images mainly include one-stage network algorithms and two-stage network algorithms [8,9,10,11,12,13,14,15,16,17]. The two-stage algorithms include R-CNN (regions with CNN features), Fast R-CNN, Faster R-CNN, Mask R-CNN, etc. These algorithms first generate the region box to be selected on the image to be detected and then perform feature extraction and classification on the image, which is slow in detection and does not meet the real-time requirements. The one-stage algorithms include SSD (Single-Shot MultiBox Detector), RetinaNet, YOLO (You Only Look Once) series, etc. These algorithms can complete the classification and regression of the anchors in one step, which improves the network detection speed but, therefore, sacrifices the detection accuracy, and the detection accuracy of the target is still lacking. Improvements in the one-stage algorithm to achieve higher detection speed and accuracy are a current research hot topic [18,19,20,21,22,23].

Miao et al. [18] used the SSD algorithm to detect insulators on transmission lines in aerial images, but only normal ceramic and composite insulators were detected, and no further insulator defect was identified. To obtain high detection accuracy of insulator defects, Jiang et al. [19] proposed an integrated multi-level perception method based on the SSD algorithm, but it takes a longer time to process the detection images. Zhao et al. [20] improved the scale-scaling module based on the STDN (Scale-Transferrable Detection Network) algorithm, thus, enabling the identification of insulators of different scale sizes, but lacking detection of insulator defects. At the same time, this network increases the number of anchors and the computational effort increases, which requires further compression of the model computation. YOLOv3 is the classic algorithm in the YOLO series, which has great advantages in model detection accuracy and detection speed. Wang et al. [21] proposed an insulator defect detection method combining a full convolutional network and YOLOv3 algorithm, and the average accuracy was significantly improved over the YOLOv3 model. However, this algorithm requires segmentation of insulators and filtering them from the background before detecting the faulty region, which is complicated. Compared with the YOLOv3 algorithm, the YOLOv4 algorithm has more advantages. Gao et al. [22] trained the detection model based on the YOLOv4 model by re-clustering the anchor box size, and the average accuracy of the model detection was improved, but the number of network layers was too great and the model was computationally complex. Han et al. [23] used an attention mechanism based on Tiny-YOLOv4 and it was tested on an embedded device with a great improvement in detection speed, but the accuracy of detection of insulators and their defects is not high. The above methods either give up real time to improve the detection accuracy of insulators and their defect or they cannot guarantee the detection accuracy, although they meet the requirements of the embedded edge end.

In order to make the target detection algorithm more suitable for deployment on edge-end devices and meet the application requirements of UAV inspection with guaranteed detection accuracy, an improved lightweight algorithm for insulator target detection and defect identification based on YOLOv4 is proposed in this paper. This algorithm improves the backbone network with a lightweight network GhostNet [24] module and embeds an attention mechanism in the feature fusion layer to improve the network’s focus on critical information. Finally, actual aerial insulator images are used for training. The improved model greatly reduces the number of parameters and the computational volume while ensuring the accuracy of insulator detection and its defect identification, with an average detection accuracy of 95.4%. The average detection accuracy is improved by 1.98% compared to YOLOv5-s and 1.29% compared to YOLOX-s. The improved model was eventually deployed on edge-side devices at a speed of 8.8 FPS, meeting the requirements for real-time target detection for UAV aerial photography.

## 2. Model Structure of YOLOv4

The YOLO series algorithm is a typical one-stage target detection algorithm, which directly performs classification and regression calculation of prediction boxes with good generalization and has wide applications in industry, transportation, medical fields, etc. Among them, the YOLOv4 algorithm model is based on YOLOv3, with improvements in data enhancement, backbone network, feature enhancement, loss function and other aspects, and can achieve an average detection accuracy improvement of about 10% on the COCO (Common Objects in Context) dataset. The YOLOv4 model mainly consists of a backbone network, a feature fusion part and a prediction layer, and its structure is shown in Figure 1.

The backbone network part is improved on the basis of YOLOv3′s Darknet by embedding five CSP (Cross Stage Partial Network) modules to form the CSPDarknet53 structure. One part of the CSP module is routinely processed for residuals and subsequently spliced directly with another part to reduce computational effort while ensuring accuracy and avoiding overfitting. The backbone part uses the Mish activation function, which is a nonlinear smoothing curve that can better transfer information to the deep network and improve the accuracy of the results. The formula of the *Mish* function is shown in Equation (1).
(1)$$Mish=x\cdot \tanh\left(\ln\left(1+e^{x}\right)\right)$$

Between the backbone network and the feature fusion part, YOLOv4 is bridged with the SPP (Spatial Pyramid Pooling) module. The SPP module performs maximum pooling at three different scales on the last feature layer of the backbone network output and then stitches together the obtained results. The SPP module can make the perceptual field of the network much larger, obtain more global information and make significant separation of important features. The feature fusion layer then uses the PANet structure to enhance feature extraction for the three initially extracted feature layers. The FPN (Feature Pyramid Network) passes down and fuses high-level information, and PANet adds a bottom-up fusion layer on top of that. In YOLOv4, PANet performs feature fusion using feature stitching to effectively fuse the high-level and underlying information fully.

The prediction part of the YOLOv4 model finally obtains three feature maps of different scales with sizes of 13 × 13, 26 × 26 and 52 × 52, and target detection is performed for the results of different sizes. Prediction judgments are made on each feature map using anchor boxes to obtain the final target detection results.

## 3. Improved YOLOv4 Insulator Target Detection and Defect Identification Algorithm

Although the detection accuracy of YOLOv4 has been improved, the YOLOv4 model is complex, with about 30,263 floating-point operations per second, which is not conducive to deployment at the edge where memory and computational resources are limited. In order to be more suitable for embedded applications while maintaining high detection accuracy, this paper proposes an improved lightweight algorithm for YOLOv4 insulator target detection and defect identification.

### 3.1. Improved Backbone Feature Extraction Network

In deep learning, overly deep network models often produce some similar feature maps, which is the key for the network to use the input information effectively. However, the generation of these similar feature maps causes an increase in computational effort. To facilitate the deployment of the model at the edge, the GhostNet network can be used to reconstruct the YOLOv4 backbone network with a simple linear transformation to produce the same rich feature maps, making it less computationally intensive and the network more lightweight [25]. The GhostNet network mainly has a Ghost Module, which is combined to build the model architecture. The structure of the Ghost Module is shown in Figure 2. First, the input feature map with a size of W×H×C is channel-reduced with normal convolution, and then additional feature maps are obtained with depthwise separable convolution. The results of normal convolution and depthwise separable convolution are superimposed to obtain a similar output feature map of size W′×H′×N. Since the normal convolution and linear transformation coexist in Ghost Module, the original features can be better preserved.

Compared with conventional convolution operations, ordinary convolution is easily limited due to its large computational size. Depthwise separable convolution is much less computationally intensive and can obtain feature maps with a more lightweight operation, so it is also often applied in lightweight networks [26]. As shown in Figure 3, the depthwise separable convolution is divided into two parts: depthwise convolution and pointwise convolution. Firstly, the number of input image channels is a. Convolution is performed on these channels separately using one convolution kernel, and there is only one convolution kernel for each channel number; the number of output channels of the obtained feature map is equal to the number of input channels, i.e., the convolution operation is performed on each channel separately. These feature maps do not integrate the corresponding feature information and need to be combined by pointwise convolution for the next step of dimensionality up or down. The resulting intermediate feature map is then convolved by 1×1×a, and the number is the number of output feature map channels b. In this way, the number of channels of the output feature map is b, which is the same as the standard convolution.

The use of depthwise separable convolution reduces the computational effort compared to normal convolution. Suppose the size of the ordinary convolutional input feature map is l×h×c, the output feature map is l′×h′×c′ and the convolutional kernel size is m×m. Then, the computational effort of ordinary convolution is $C_{1}=l\prime \times h\prime \times c\prime \times m\times m\times c$. In the Ghost Module, the convolution kernel size is assumed to be $m_{1}$ for the first normal convolution and $m_{2}$ for depthwise separable convolution, generating n additional feature maps. It is known that Ghost Module compresses the calculated amount to $\frac{1}{n}$ of the original amount.

As shown in Figure 4, two Ghost Modules form Ghost Bottlenecks, with the first Ghost Module up-dimensioning the channels and the second Ghost Module down-dimensioning them to facilitate the connection of the input feature maps. Consider the cases stride = 1 and stride = 2.

### 3.2. Incorporating Lightweight Attention Mechanism Module

The use of lightweight networks reduces the number of computational parameters and the complexity of the algorithm model, but at the same time, causes the extraction of effective feature information that is not rich enough, which easily causes a decrease in detection accuracy. To improve the model’s focus on the detection target features, this paper incorporates an attention mechanism in the network. In computer vision, the attention mechanism enables the model to acquire more useful information and focus on the desired target region to enhance the feature extraction capability of the network [27,28]. Among them, the channel attention mechanism allows the network to focus on the important channels in the input image and continuously deepen the information of features on different channels, which facilitates the model to learn the feature information of the detection target and, thus, locate that target, such as SE-Net (Squeeze-and-Excitation Networks) [29], ECA-Net (Efficient Channel Attention Neural Networks), etc. Spatial attention mechanisms complete learning and augmentation of pixel information of spatial locations, focusing on locating key location information, allowing models to find target locations more accurately, such as Non-Local, STN (Spatial Transformer Networks), etc. A hybrid attention mechanism is a combination of both, which needs to coordinate spatial attention and channel attention but may generate redundant information, such as CBAM (Convolutional Block Attention Module), Coordinate Attention, etc. Among these attention mechanisms, the smaller modules are ECA-Net, Non-Local, CBAM, etc. ECA-Net belongs to the modules with a relatively small number of operations to obtain cross-channel information in an efficient way.

Therefore, to learn the features of insulators and their defect while ensuring overall light weight, this paper introduces a lightweight channel attention mechanism, ECA-Net, which makes the network extract the more important information in the input image using a small number of parameters [30]. ECA-Net can be seen as an improved SE-Net. SE-Net is a typical channel attention mechanism that captures the different weights of different channels and makes the network pay more attention to the important channels. The structure is shown in Figure 5, where W, H and C denote the length, width and the number of channels of the feature map, respectively, GAP denotes global average pooling, FC denotes fully connected and sigmoid denotes the sigmoid function.

The SE module obtains the weights of each channel via Formula (2):(2)$$\omega_{1}=\sigma \left(\mu_{2}r\left(\mu_{1}z\right)\right)$$
where z denotes the result of global averaging pooling of the input features, $\mu_{1}=C\times \frac{C}{r}$ $\mu_{2}=\frac{C}{r}\times C$, σ is the Sigmoid function and r is the ReLu function.

The ECA module is based on the SE module. It avoids the complex correlation of different channels brought about by the dimensionality reduction in channel attention. Its structure is shown in Figure 6. After global average pooling, a one-dimensional convolutional (Conv1D) kernel of size *k* is used instead of the fully connected layer to achieve cross-channel interaction between channels and obtain the weights of different channels in the feature map. The weights are multiplied with the input feature map to obtain the output feature map.

The ECA module obtains the weights of each channel via Formula (3):(3)$$\omega_{2}=\sigma \left(C1D_{k}\left(z\right)\right)$$

The convolution kernel size *k* is calculated by Equation (4):(4)$$k=\psi (C)=\frac{\log_{2}\left(C\right)+1}{2}$$

Since the weights obtained by the SE module correspond to each channel indirectly, its attention effect is not optimal. In contrast, the ECA module completes the information interaction between the channels locally, which helps to improve the efficiency of the model to capture the attention of the channels.

The attention mechanism can be embedded in different locations in the model, and different effects may occur in different parts. In this paper, we propose two fusion methods: fusing the ECA module into the Ghost Bottleneck module of the backbone network or the feature fusion stage of the model to further adjust the model and improve its ability to extract important information.

As an example, when stride = 1 in the Ghost Bottleneck module, the ECA module is embedded in the Ghost Bottleneck module after the first Ghost Module, and the improved ECA–Bottleneck module is used to form the ECA–GhostNet backbone network to improve the feature extraction capability of the network. The structure of the improved ECA–Bottleneck module is shown in Figure 7.

2.In the feature fusion part, the ECA module can be considered to be embedded after the three feature layers have already been extracted to further improve the attention of the model to the feature information. The attention mechanism is also embedded after the up-sampling of PANet to enhance the global information fusion and make the interaction of model contextual information more effective.

### 3.3. Improved YOLOv4 Algorithm Model

The improved YOLOv4–GhostNet–ECA structure is shown in Figure 8. The backbone network of YOLOv4 is reconstructed into the GhostNet network, and the feature fusion stage is improved by depthwise separable convolution, resulting in a reduced number of model parameters. To ensure the accuracy of detection results and enhance the feature extraction capability, the ECA-Net attention mechanism is embedded after the three effective feature layers are extracted by the backbone network. The same ECA-Net attention mechanism is also embedded in PANet to enhance the local information extraction capability after up-sampling. The final prediction was obtained by YOLO Head.

## 4. Experimental Results and Analysis

### 4.1. Dataset and Experimental Environment

The model training selection is a PyTorch deep learning framework with CUDA = 11.2 and the platform configuration is Intel Xeon Platinum 8171 M@ 2.60 GHz CPU with 6 × 16 GB RAM and NVIDIA RTXA6000 graphics card configuration. The configuration for the test on the local computer is Windows 11, NVIDIA GeForce GTX 3060 GPU.

In the training process, the model may have the problem of good effect on the training set but poor generalization effect on the test set, which is called over-fitting. After data expansion, richer datasets can be used in training, thus, reducing over-fitting. The insulator dataset used in the experiment was collected from the aerial images of the UAV site and the images were pre-processed. After rotating, stitching, flipping, adjusting the brightness and other operations to expand the samples, there were 1588 insulator images. In view of the redundant data that may be produced by aerial images, this paper removes the low-quality images with serious occlusion and inconspicuous features, so as not to affect the training results. At the same time, some defects are appropriately added to reduce sample imbalance. Labeling software was used to label the dataset images, and the locations of insulators and defects in the images were labeled in the VOC dataset format, noted as insulator and defect, respectively. Some of the insulators with defect images are shown in Figure 9, with red boxes marked as insulators and yellow boxes marked as defects.

Generally, datasets are divided into training set, validation set and test set. The training set can be used to train the model, which is convenient to adjust the parameters. When the model is updated with different parameters, the effect of the model is evaluated on the validation set to continuously improve the stability of the model. Finally, the generalization effect of the model is obtained on the test set. Randomly dividing the dataset can ensure the uniform distribution of image samples, prevent a small number of similar images taken in repeated positions from being trained, be beneficial to model training and evaluation and reduce over-fitting. Therefore, the dataset is randomly divided into a ratio of 8:1:1, which contains 1286 images for the training set, 143 images for the validation set and the remaining 159 images for the test set. It is convenient to train with the improved lightweight YOLO4 network afterward.

### 4.2. Experimental Procedure

In the training process, pre-trained weight obtained by training on large datasets is used in conjunction with migration learning. The training is first frozen for 50 rounds to freeze the backbone feature extraction network and speed up the network training. The learning rate is set to 0.001 and the batch size is set to 16. Subsequently, 150 rounds of training were unfrozen, the learning rate was adjusted to 0.0001, the batch size was set to 8 and, in total, 200 rounds were trained. The size of batch_size will affect the data processing effect of Batch Normalization. As a normalization method, Batch Normalization can accelerate the convergence speed of the network in the training process, improve the generalization ability of the model and avoid over-fitting. Since too-deep training will lead to the lack of learning of common laws in the model, 200 rounds of training are chosen to prevent over-fitting.

Since the anchor size of the original YOLOv4 network was not suitable for the insulator dataset, nine anchors were regenerated using the K-means clustering algorithm with sizes of (22,23), (48,21), (22,51), (88,31), (42,101), (116,48), (270,78), (99,280) and (279,128) to make it easier for the anchors to match the target features, which were used on the feature maps on three scales: large, medium and small.

Figure 10 shows the training loss profiles of the YOLOv4–GhostNet network and the YOLOv4–GhostNet network after embedding the ECA attention mechanism; the horizontal coordinate is the number of network iterations and the vertical coordinate is the loss value. It can be seen that the net loss is stable after 150 rounds of training, and the net loss is smaller after embedding the ECA-Net attention mechanism.

### 4.3. Evaluation Indicators

To evaluate the target detection network performance, the experimental results were analyzed and evaluated with the appropriate metrics. The model strengths and weaknesses were chosen to be measured in terms of the average precision (AP) for each category of detected targets, the average precision for all categories (mAP), the model detection speed, i.e., the number of frames per second FPS, the model size and the miss detection rate, where AP is the area under the precision and recall curves, representing the average of the model under different recall rates, which can be used to more comprehensively assess how good the model is. The higher the AP, the more accurate the model identifies the target; the larger the FPS, the faster the model detects; the smaller the miss detection rate, the fewer targets are not detected. The formulae for calculating the accuracy, recall and miss detection rates are as follows.
(5)$$P_{re}=\frac{TP}{TP+FP}$$
(6)$$R_{e}=\frac{TP}{TP+FN}$$
(7)$$M_{r}=\frac{FN}{TP+FN}$$

*TP* means that the prediction is correct for a positive sample and is actually positive as well; *FP* means the prediction is a positive sample, but the prediction is wrong and the actual sample is negative; *FN* indicates a negative predicted sample and a positive actual sample, i.e., a positive sample that is not detected.

*AP* and *mAP* are calculated as follows. There are *n* categories in total.
(8)$$AP=\int_{0}^{1}P\left(r\right)dr$$
(9)$$mAP=\frac{\sum_{i=1}^{n}AP_{i}}{n}$$

### 4.4. Comparison and Analysis of Results

To verify the effectiveness of incorporating the ECA-Net attention mechanism, the results of the two improved model structures were compared and analyzed with those of the original YOLOv4 model, as shown in Table 1.

As can be seen from Table 1, the average detection accuracy of the original YOLOv4 for both insulator and insulator defect targets is 94.63%. After reconfiguring the backbone network of YOLOv4 into the lightweight GhostNet, the average detection accuracy decreases slightly to about 0.7%, in which the detection accuracy for insulator defect decreases by about 1.3%, but the model complexity achieves a significant reduction, the model size is only 17.4% of the original one and the detection speed is increased from 41 FPS to 74 FPS, which meets the requirements of embedded edge-end devices.

The average detection accuracy is not significantly improved after incorporating the Ghost Bottleneck of the backbone network GhostNet into the ECA-Net attention mechanism compared to improving only the backbone network, indicating that the ECA-Net attention mechanism does not focus more useful information in the feature extraction stage of the network. Based on the YOLOv4–GhostNet network, the ECA-Net attention mechanism is incorporated in the feature fusion stage, and the average detection accuracy of the model increases by about 1.5% compared with that without the attention mechanism, especially the recognition accuracy of insulator defect, which is improved by nearly 3.5%, indicating that the ECA-Net attention mechanism is useful for improving the target attention and extracting more required feature information is more helpful. Therefore, this paper finally chooses to embed the attention mechanism into the feature fusion part.

The results of comparing the algorithm in this paper with embedding other lightweight attention mechanisms are shown in Table 2.

As can be seen from Table 2, the average detection accuracy is only 93.70% when using the spatial attention mechanism Non-Local. When using the hybrid attention mechanism CBAM, the average detection accuracy is 94.33%. However, due to the complex structure of CBAM, the channel features need to be extracted first and then the weights of the spatial domain are obtained, and the detection speed is only 60 FPS. The YOLOv4–GhostNet–ECA algorithm achieves a balance between detection accuracy and detection speed and achieves 95.65% detection accuracy for insulator defects while maintaining a detection speed of 70 FPS, indicating that ECA-Net can effectively improve the model’s ability to detect small targets.

To ensure the reliability of this algorithm when deployed in edge-end devices, the algorithm in this paper is compared with Tiny-YOLOv4, improved Tiny-YOLOv4 [23] and the lightweight YOLOv5-s, YOLOX-s algorithms and other mainstream algorithms, as shown in Table 3.

From Table 3, it can be seen that Faster R-CNN and SSD algorithms cannot meet the detection of insulator defects; Tiny-YOLOv4 has the fastest detection speed, but due to the simple structure, the detection accuracy is sacrificed, and the average detection accuracy is only 89.14%, especially the insulator defect recognition accuracy, which is only 85.43% and cannot well meet the requirements of target detection and defect recognition. The algorithm proposed in the literature [23] effectively improves the detection accuracy of insulator defects, but the average detection accuracy is still low; the model size of the YOLOv5-s and YOLOX-s algorithms is reduced compared with the improved algorithm in this paper, but the detection accuracy of insulator defect is reduced in both; the algorithm in this paper can guarantee a high detection accuracy and meet the real-time requirements at the same time.

Figure 11 shows a comparison of the detection results using the original YOLOv4 algorithm and the YOLOv4–GhostNet–ECA algorithm proposed in this paper in different scenarios. Figure 11a shows that the original YOLOv4 misses the detection of small targets such as insulator defects and fails to detect overlapping insulators when there are overlapping targets. The original YOLOv4 network misses detection when there are multiple insulators. The detection effect of the YOLOv4–GhostNet–ECA algorithm is shown in Figure 11b, which can detect insulator defects and overlapping insulators. When there are multiple insulators, the algorithm in this paper can detect seven insulators. Among them, the $M_{r}$ of the YOLOv4 algorithm is 3.2% for insulators and 9.1% for broken insulators; the $M_{r}$ of this paper’s algorithm is 2.3% for insulators and 5.2% for broken insulators, which better reduces the miss detection rate of the target.

In order to verify the effectiveness of the ECA-Net attention mechanism for improving the attentional goals of the network, a heat map visualization was chosen for the analysis. The brighter regions in the graph represent the areas where the network pays more attention. As shown in Figure 12a, the YOLOv4–ghostnet algorithm does not pay enough attention to insulator defects has missed the detection of smaller insulators. In the complex background, only three insulators can be detected. Figure 12b shows that the YOLOv4–GhostNet–ECA algorithm can focus on defects more accurately and detect insulators in overlapping cases. Five insulators can be detected in the complex background.

### 4.5. Hardware Device Deployment Test

The algorithms in this paper are ported to Jetson Xavier NX edge devices, and the improved YOLOv4–GhostNet–ECA model is verified to be suitable for deploying edge-side devices by invoking real-time camera detection. Jetson Xavier NX can provide a good operating environment for AI models on embedded devices and facilitate UAV development and deployment.

Table 4 shows the comparison of the operation results of other models and this paper’s algorithm in edge-end devices. From Table 4, we can see that the FPS of the YOLOv5-s model is 10.5, but the detection accuracy is not as good as the algorithm in this paper. The FPS of the YOLOX-s model is 6.8 and the detection speed is slower. The improved YOLOv4–GhostNet–ECA model doubles the detection speed compared to the original YOLOv4 model, reaching 8.8 FPS, which meets the UAV aerial photography detection requirements. 

Figure 13a shows the embedded device unit; Figure 13b shows the detection effect of running this paper’s algorithm on a Jetson Xavier NX device.

## 5. Conclusions

This paper proposes a lightweight insulator target detection and defect-recognition algorithm based on improved YOLOv4, which reconfigures the backbone network in the YOLOv4 network from DarkNet53 to GhostNet and extracts the features of input images with a more lightweight approach. In the feature fusion stage, depthwise separable convolution is used to improve the efficiency of network calculation; in order to ensure that important information can be identified, the ECA-Net attention mechanism is embedded. Comparing the proposed algorithm with the original YOLOv4, the experiment shows that mAP is improved from 94.63% to 95.40%. The model size is compressed from 244 MB to 42 MB, and the detection speed is increased from 41 FPS to 70 FPS. Tested on Jetson Xavier NX equipment, the detection speed is 8.8 FPS, which is convenient for the deployment of edge devices. In this paper, the algorithm meets the requirements of accuracy and speed in detecting insulator defects by UAV, and the network is lightweight. Later, the identification of other faults, such as pollution and flashover burns, will be further studied.

## Figures and Tables

**Figure 1 sensors-23-01216-f001:**
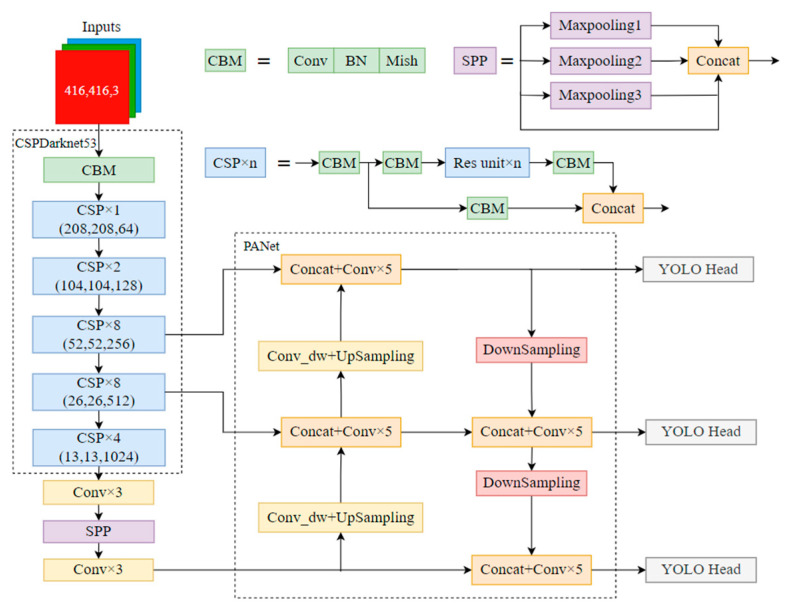
Structure of the YOLOv4 model.

**Figure 2 sensors-23-01216-f002:**
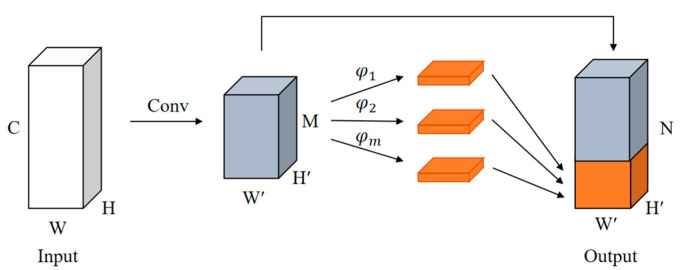
Structure of the Ghost Module.

**Figure 3 sensors-23-01216-f003:**
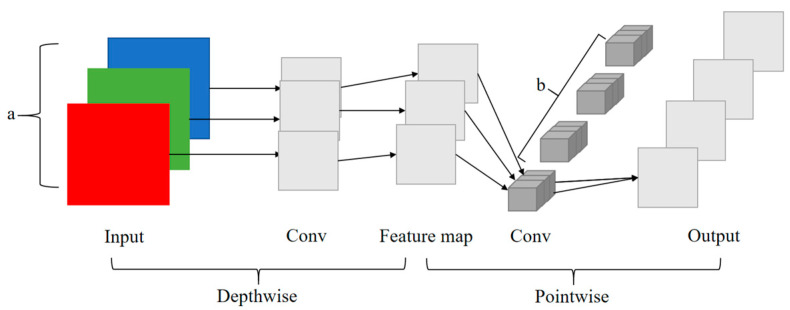
Depthwise separable convolution.

**Figure 4 sensors-23-01216-f004:**
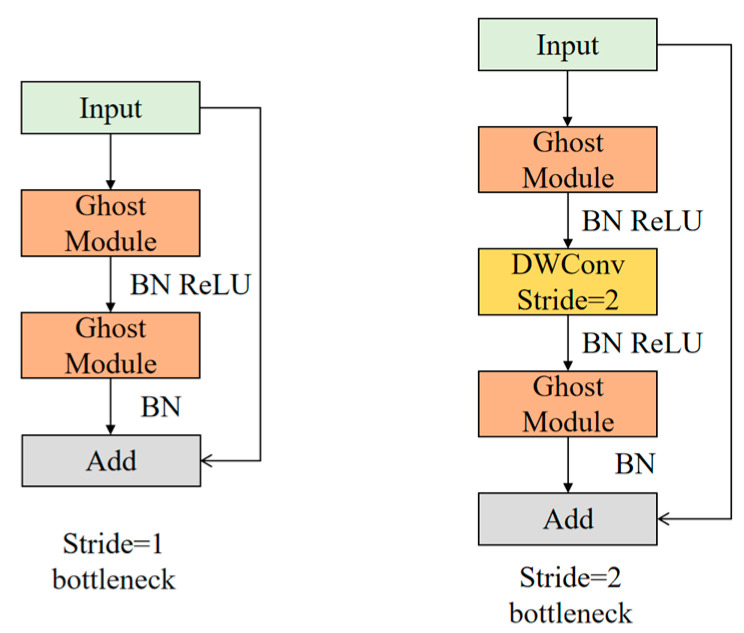
Structure of Ghost Bottlenecks.

**Figure 5 sensors-23-01216-f005:**
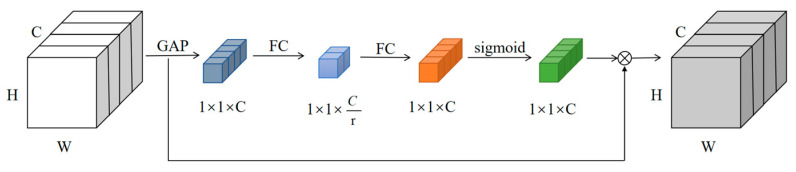
Structure of the SE module.

**Figure 6 sensors-23-01216-f006:**
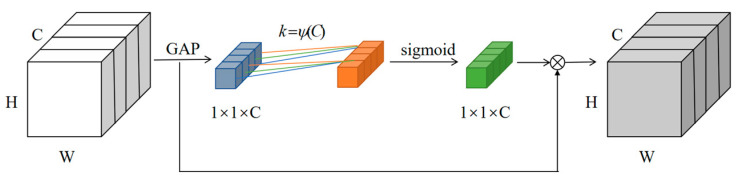
Structure of the ECA module.

**Figure 7 sensors-23-01216-f007:**
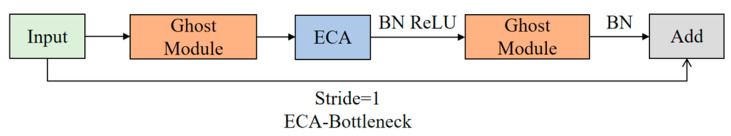
Improved ECA–Bottleneck module.

**Figure 8 sensors-23-01216-f008:**
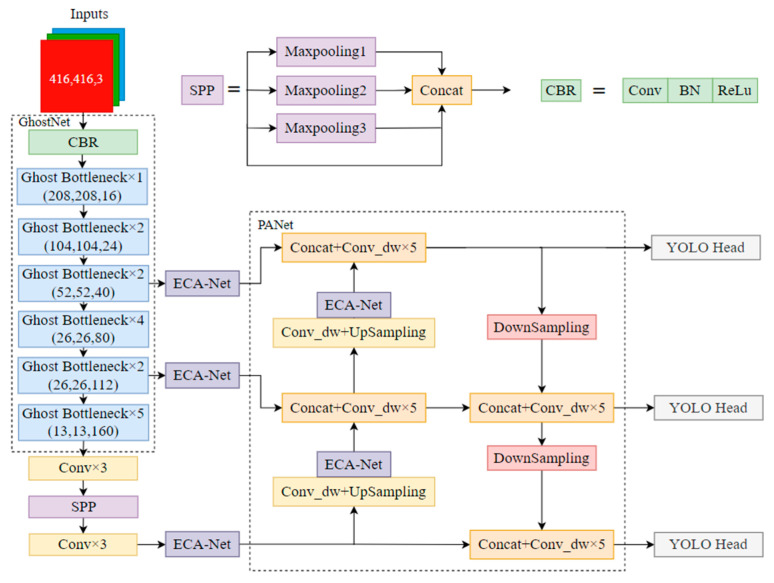
Improved YOLOv4 network structure.

**Figure 9 sensors-23-01216-f009:**
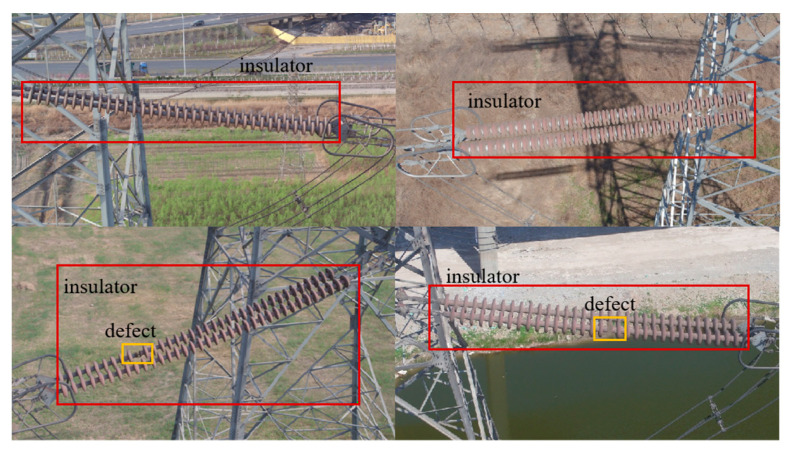
Insulator and defect images.

**Figure 10 sensors-23-01216-f010:**
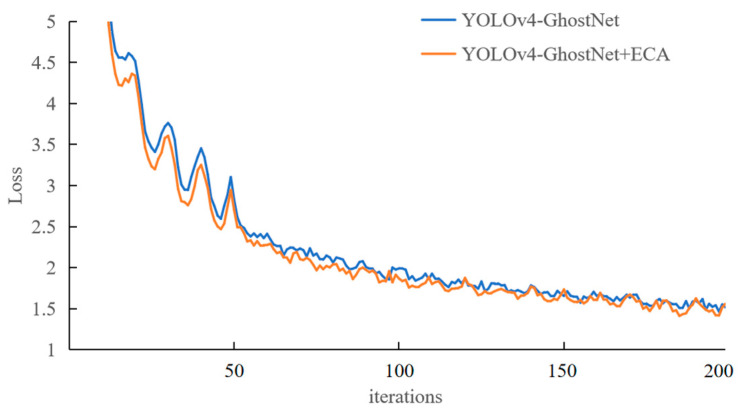
Network training loss graph.

**Figure 11 sensors-23-01216-f011:**
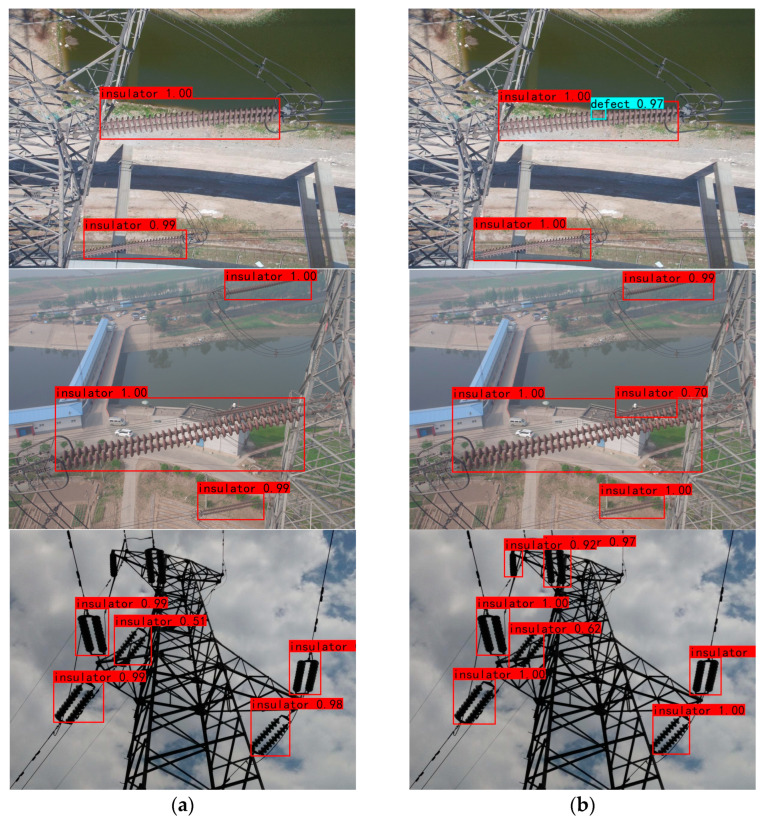
Comparison of the detection effects: (**a**) YOLOv4 algorithm; (**b**) YOLOv4–GhostNet–ECA algorithm.

**Figure 12 sensors-23-01216-f012:**
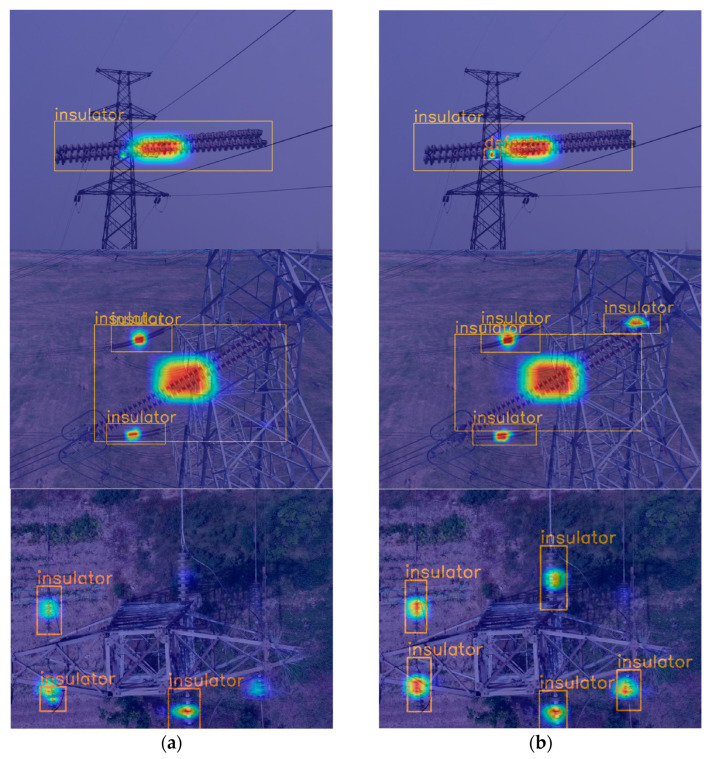
Heat map visualization: (**a**) YOLOv4–GhostNet algorithm; (**b**) YOLOv4–GhostNet–ECA algorithm.

**Figure 13 sensors-23-01216-f013:**
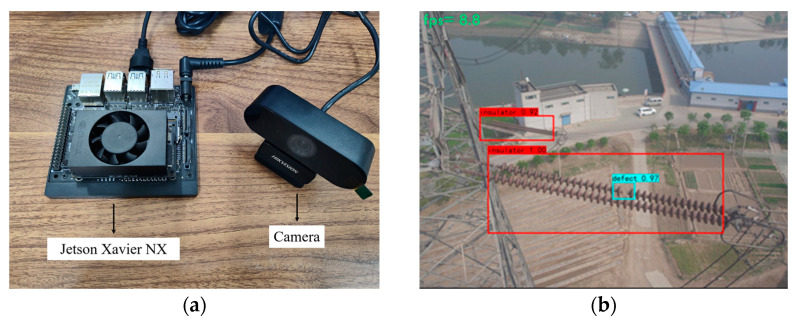
Embedded devices and deployment results: (**a**) the embedded device unit; (**b**) the effect of edge-end device operation.

**Table 1 sensors-23-01216-t001:** Comparison of the effects of improved attention mechanisms.

Model	mAP (%)	Insulator AP (%)	Defect AP (%)	Size (MB)	FPS
YOLOv4	94.63	95.83	93.44	244.32	41
YOLOv4–GhostNet	93.95	95.76	92.13	42.40	74
YOLOv4-ECA-Bottleneck	94.76	95.35	94.17	42.50	68
YOLOv4–GhostNet–ECA	95.40	95.15	95.65	42.40	70

**Table 2 sensors-23-01216-t002:** Comparison of cross-sectional effects of attentional mechanisms.

Model	mAP (%)	Insulator AP (%)	Defect AP (%)	Size (MB)	FPS
Non-Local	93.70	95.26	92.14	42.60	68
CBAM	94.33	95.28	93.37	42.60	60
ECA	95.40	95.15	95.65	42.40	70

**Table 3 sensors-23-01216-t003:** Comparison of different models.

Model	mAP (%)	Insulator AP (%)	Defect AP (%)	Size (MB)	FPS
Faster R-CNN	85.74	95.82	75.67	521.00	21
SSD	83.77	91.96	75.59	91.10	77
Tiny-YOLOv4	89.14	92.85	85.43	22.46	149
Han et al. [23]	92.69	93.97	91.42	24.97	121
YOLOv5-s	93.42	96.66	90.18	27.10	83
YOLOX-s	94.11	96.35	91.88	34.30	67
YOLOv4–GhostNet–ECA	95.40	95.15	95.65	42.40	70

**Table 4 sensors-23-01216-t004:** Results of edge-end device operation.

Model	mAP (%)	Size (MB)	FPS
YOLOv4	94.63	244.32	4.5
YOLOv5-s	93.42	27.10	10.5
YOLOX-s	94.11	34.30	6.8
YOLOv4–GhostNet–ECA	95.40	42.40	8.8

## Data Availability

Dataset link: https://github.com/InsulatorData/InsulatorDataSet (accessed on 18 May 2022).

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
