# Peer review of "A Lightweight Algorithm for Insulator Target Detection and Defect Identification"

_sensors, 2023, doi:10.3390/s23031216_

Round 1

Reviewer 1 Report

This research proposes a lightweight insulator target detection and defect-recognition algorithm based on improved YOLOv4, which reconfigures the backbone network in the YOLOv4 network from DarkNet53 to GhostNet and extracts the features of input images with a more lightweight approach. Comparing the trained algorithm of this paper with the original YOLOv4, it can be seen experimentally that the average accuracy of the improved network detection is improved to 95.40%, which is 1.98% and 1.29% better than the YOLOv5-s and YOLOX-s models respectively. The model size is compressed from 244MB to 42MB, the detection speed is increased from 41 FPS to 70 FPS.

The algorithm in this paper improves the accuracy and speed of UAV in insulator defect detection, and can better meet the actual needs of the project. However, there are still some problems in this article, please modify it.

Peer review 1:

In this paper, there is a common problem that the distance between the picture and the text above is too narrow, so the spacing should be widened appropriately.

Peer review 2:

In this paper, there are some formatting errors, which make the paper unprofessional. For example, the edited part of the formula editor in the article is not on the same level as the text, the title of Table 4 is not on the same page as the main part.

Peer review 3:

Page 11 of the article shows the comparison of the ability of YOLOv4 algorithm and YOLOv4-ghostneteca algorithm to identify defective insulators, but the number of verification sets is not marked, please explain the number of verification sets.

Peer review 4

The abstract and conclusion part of the article is too long, please simplify it so as to highlight the core work of the paper.

Author Response

Dear Reviewer,

We are very grateful for your professional comments on our article. According to your suggestion, we have made extensive corrections to the previous manuscript, and the specific corrections are as follows. Please see the attachment.

Point 1: In this paper, there is a common problem that the distance between the picture and the text above is too narrow, so the spacing should be widened appropriately.

Response 1: Thanks for your careful checks. We are sorry for our carelessness. According to your opinion, we have appropriately increased the spacing between words and pictures.

Point 2: In this paper, there are some formatting errors, which make the paper unprofessional. For example, the edited part of the formula editor in the article is not on the same level as the text, the title of Table 4 is not on the same page as the main part.

Response 2: Thank you for your correction. Due to our negligence, we didn't adopt the correct format. We have adjusted the format of the article, such as lines 173-177. The format of the table has also been adjusted.

Point 3: Page 11 of the article shows the comparison of the ability of YOLOv4 algorithm and YOLOv4-ghostneteca algorithm to identify defective insulators, but the number of verification sets is not marked, please explain the number of verification sets.

Response 3: As suggested by the reviewer, We marked the number of data set partitions in lines 304-308 of the article. The following are the specific proportions.

The dataset is randomly divided in the ratio of 8:1:1, which contains 1286 images for the training set, 143 images for the validation set, and the remaining 159 images for the test set. It is convenient to train with the improved lightweight YOLO4 network afterward.

Point 4: The abstract and conclusion part of the article is too long, please simplify it so as to highlight the core work of the paper.

Response 4: We have re-written this part according to the Reviewer’s suggestion. In the revised edition, all changes to our manuscript are highlighted in red text in the document.

We would like to thank the reviewer again for taking the time to review our manuscript. Thank you very much for your comments and suggestions.

Reviewer 2 Report

Authors have worked on the application of image capturing processing through an improved YOLOv4 approach. Authors claim to have achieved a high detection accuracy with an appreciable fps speed. Authors validated their approach with already existing approaches in the literature. Performance of the proposed approach was compared in terms of mAP(%), Insulator AP(%), Defect AP(%), Size(MB) and FPS. Work seems interesting and of a very matured quality. However, I have few minor suggestions.

1. Authors are encouraged to write more on how they have reduced overfit without compromising accuracy.

2. Images have been captured aerially. Did the authors pre-process the data in terms of redundancy as well. 

3. Why is there a random division for the training, validation and test sets? Can there be an decisive and informed division? If No, why and if yes, how? Data pre-processing and filtering is one of the most challenging aspect in application of ML/data techniques and should be clarified for benefit of the readers.

Author Response

Dear Reviewer,

We sincerely thank the reviewer for their valuable feedback, which we use to improve the quality of the manuscript. The reviewer's comments are listed below, and the specific questions have been numbered. Our reply is given in red text. Please see the attachment.

Point 1: Authors are encouraged to write more on how they have reduced overfit without compromising accuracy. 

Response 1: Thanks for your suggestion. The method to reduce over-fitting can start with the data set, such as expanding the data set and training with rich samples. During the experiment, adopting Batch Normalization and early stopping can also reduce the risk of over-fitting. We have added relevant explanations in sections 4.1 and 4.2 of the manuscript. The following is a specific explanation.

In the training process, the model may have the problem of good effect on the training set but poor generalization effect on the test set, which is called over-fitting. After data expansion, richer data sets can be used in training, thus reducing over-fitting.

As a normalization method, Batch Normalization can accelerate the convergence speed of the network in the training process, improve the generalization ability of the model and avoid over-fitting. Since too deep training will lead to the lack of learning of com-mon laws in the model, 200 rounds of training are chosen to prevent over-fitting.

Point 2: Images have been captured aerially. Did the authors pre-process the data in terms of redundancy as well.

Response 2: As suggested by the reviewer, We have supplemented the preprocessing of redundant data. The insulator images obtained by aerial photography are easy to be blocked and similar. In lines 289-292, we introduce how to pre-process them. The specific details are as follows.

In view of the redundant data that may be produced by aerial images, this paper re-moves the low-quality images with serious occlusion and inconspicuous features, so as not to affect the training results. At the same time, some defects are appropriately added to reduce sample imbalance.

Point 3: Why is there a random division for the training, validation and test sets? Can there be an decisive and informed division? If No, why and if yes, how? Data pre-processing and filtering is one of the most challenging aspect in application of ML/data techniques and should be clarified for benefit of the readers.

Response 3: Thanks for your suggestion. In the initial experiment, we used a variety of different proportions of data set division methods. Through experiments, the effect of randomly dividing data sets with the ratio of 8: 1: 1 is relatively optimal, so this ratio is adopted in subsequent experiments. According to this ratio, the image data for training are sent in random order, not in file position order. The main reason is that a small number of images in similar positions have repeated shooting, so they are trained in random order. On the other hand, over-fitting can also be reduced. The contents of this part are revised in lines 297-307, as follows.

Generally, data sets are divided into training set, validation set and test set. The training set can be used to train the model, which is convenient to adjust the parameters. When the model is updated with different parameters, the effect of the model is eval-uated on the validation set to continuously improve the stability of the model. Finally, the generalization effect of the model is obtained on the test set. Randomly dividing the data set can ensure the uniform distribution of image samples, prevent a small number of similar images taken in repeated positions from being trained, be beneficial to model training and evaluation, and reduce over-fitting. Therefore, the dataset is randomly di-vided in the ratio of 8:1:1, which contains 1286 images for the training set, 143 images for the validation set, and the remaining 159 images for the test set. It is convenient to train with the improved lightweight YOLO4 network afterward.

We would like to thank the referee again for taking the time to review our manuscript.We sincerely thank the reviewer for your work and hope that the revision can be recognized. Thank you again for your comments and suggestions.
